# PIMT: Physics-Based Interactive Motion Transition for Hybrid Character Animation

Yanbin Deng
202222280715@std.uestc.edu.cn
University of Electronic Science and Technology of China
Chengdu, Sichuan, China

Zheng Li
202152080128@std.uestc.edu.cn
University of Electronic Science and Technology of China
Chengdu, Sichuan, China

Ning Xie[*]
seanxiening@gmail.com
University of Electronic Science and Technology of China
Chengdu, Sichuan, China

Wei Zhang
37058836@qq.com
University of Electronic Science and Technology of China
Chengdu, Sichuan, China

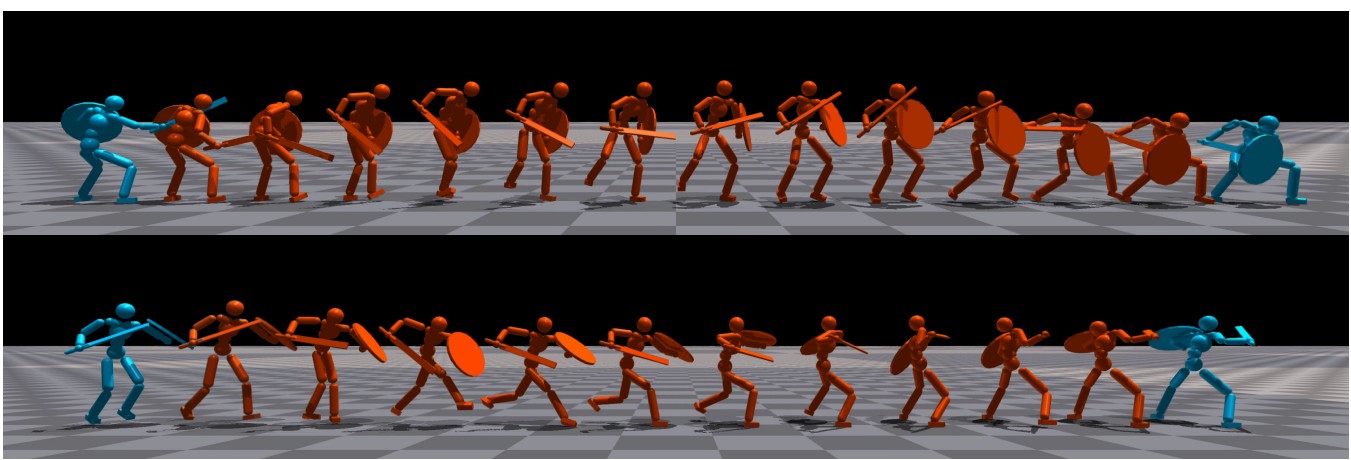

**Figure 1: Our framework can handle diverse interactive transition tasks with high accuracy and naturalness. Here, the policy controls the character to generate transition movements (in orange) between the source and target poses (in blue), thus smoothly bridging the front and back motion clips (top: stab and shield bash; bottom: run and slash).**

## Abstract

Motion transitions, which serve as bridges between two sequences of character animation, play a crucial role in creating long variable animation for real-time 3D interactive applications. In this paper, we present a framework to produce hybrid character animation, which combines motion capture animation and physical simulation animation that seamlessly connects the front and back motion clips. In contrast to previous works using interpolation for transition, our physics-based approach inherently ensures physical validity, and both the transition moment of the source motion clip and the

horizontal rotation of the target motion clip can be specified arbitrarily within a certain range, which achieves high responsiveness and wide latitude for user control. The control policy of character can be trained automatically using only the motion capture data that requires transition, and is enhanced by our proposed Self-Behavior Cloning (SBC), an approach to improve the unsupervised reinforcement learning of motion transition. We show that our framework can accomplish the interactive transition tasks from a fully-connected state machine constructed from nine motion clips with high accuracy and naturalness.

## CCS Concepts

• **Computing methodologies** → **Procedural animation**; *Physical simulation*; *Reinforcement learning*; Motion capture.

## Keywords

Character Animation, Unsupervised Reinforcement Learning, Physics-Based Simulation and Control, State Machine

**ACM Reference Format:**
Yanbin Deng, Zheng Li, Ning Xie, and Wei Zhang. 2024. PIMT: Physics-Based Interactive Motion Transition for Hybrid Character Animation. In *Proceedings of the 32nd ACM International Conference on Multimedia (MM*

[*]Corresponding author

'24), October 28-November 1, 2024, Melbourne, VIC, Australia. ACM, New York, NY, USA, 9 pages. https://doi.org/10.1145/3664647.3681582

## 1 Introduction

In Real-time 3D interactive applications such as video games, humanoid characters are required to perform corresponding motions according to the instructions input by the user. Currently, the standard and well-established practice is to use motion capture animation organized by a state machine[5, 31], and one of the key research topics is to achieve a natural transition of two motion clips. For this issue, early practices include letting all motion clips start and stop in a uniform idle pose or creating transition motion clips for every possible combination of transition[32]. The previous approach has implications for the naturalness of transition, since a transfer posture was forcibly assigned to it. As for the latter method, it is not trivial for animators to create transitions between two arbitrary motion clips, especially when the amount of motion clips increases. Furthermore, the transition points in both methods can only be fixedly specified by the animator and, therefore, cannot respond to user inputs at an arbitrary frame.

By far, the most common solution for motion transition in the game industry remains interpolation [3, 4, 13, 33] for its simplicity and computational efficiency. In this approach, transitions are generated by blending two motion clips to create a visually compelling and seamless motion. This solution can be applied to interactive motion transition. However, the motions it generates are not constrained by the human body dynamics model and thus may produce severe visual artifacts when the difference between source and target poses is huge. On the other hand, the animation generated by physics-based character control [22, 23, 34, 35] inherently guarantees physical validity. However, They are not designed to create transitions between two motion clips. Most of these works require imitation of large reference motion datasets to ensure naturalness; when dealing with interactive motion transition, they may suffer from the mismatch between the motions from the training dataset and the desired transition motions.

In this work, we aim to develop a framework to generate long term character animation with the hybrid pattern [26] interactively, which leverages the physics-based character control to bridge the motion clips seamlessly, and both the transition moment and the target motion rotation can be specified in real-time. As far as we know, previous researchers have yet to investigate such an implementation of interactive motion transition. The framework only utilizes the motion clips that require transition for the unsupervised reinforcement learning, and our proposed SBC mechanism will further utilize the knowledge of transition in the exploration process. The SBC provides animators with greater creative freedom, as they do not need to ensure the similarity of the motions in their creation to the reference dataset. Previous works of hybrid animation often require blending post-processing to align the physics-based animation with the motion clip [36], while our control policy trained with sophisticated reward function and curriculum learning strategy can align with target pose very precisely, in the meantime eliminating several noticeable visual artifacts in the motion. Our contributions can be summarized as follows:

- We propose a novel hybrid framework to archive interactive motion transition and design a sophisticated reward function that integrates transition accuracy and naturalness objectives for unsupervised reinforcement learning. The well-trained policy considers accuracy, naturalness, rapidity, and robustness.
- We introduce a task planning mechanism with a curriculum learning strategy; thus the policy can be trained automatically with the designer-defined state machine and efficiently achieve the highest accuracy.
- We present Self-Behavior Cloning (SBC) to help the policy better utilize its exploration trajectories, thereby learning richer knowledge of motion transition and improving the result.

## 2 Related Work

### 2.1 Interpolation

Interpolation techniques, introduced by [24], use ease-in-ease-out to blend two motion clips smoothly, and the source motion clip fades out as the target motion clip fades in. Time-warping technology introduced by [3] is commonly used in these works, which align two motion sequences by stretching or compressing them in time. Several previous interpolation-based methods specifically focused on interactive motion transition, which requires responsiveness to user input and efficiency to run in real-time. Egbert et al. [4] use Laplacian pyramid decomposition to keep the features of original motion; Ikemoto et al. [10] precompute a look-up table for the weights of multi-way blends and recover the blend recipe at runtime; Koyama et al. [13] search for an optimal intermediate motion that minimizes the duration for the transition while maximizing the naturalness.

Compared with our physics-based method, interpolation-based approaches inherently distort the actual motion and do not consider the dynamic model of the human body. Thus the generated transitions may not be realistic. In addition, most of these methods used a fixed duration for the transition. Meanwhile, ours let the control policy decide the duration according to the difficulty of transition (similar to [13]), which is consistent with real human movement.

### 2.2 Motion Graphs

In this approach, a set of motion capture data is structured as a motion graph, whose edges represent either pieces of original motion clips or automatically generated transitions [12], i.e., the blends between poses that are similar enough. Many works studied the criteria for selecting appropriate nodes [2, 14, 30, 33], i.e. rational transition points of motion clips. After the graph creation, a graph search technique obtains the motion transition.

Some variations of motion graphs have been proposed for better adaptation to interactive applications [6, 14, 15]. However, a common problem with this approach is its insufficient responsiveness, as noted by [13], since the path from the source motion to the target motion on a graph can be unacceptably long. In contrast, our approach uses a designer-defined state machine to organize animation intuitively, and the well-trained control policy can ensure a swift transition.

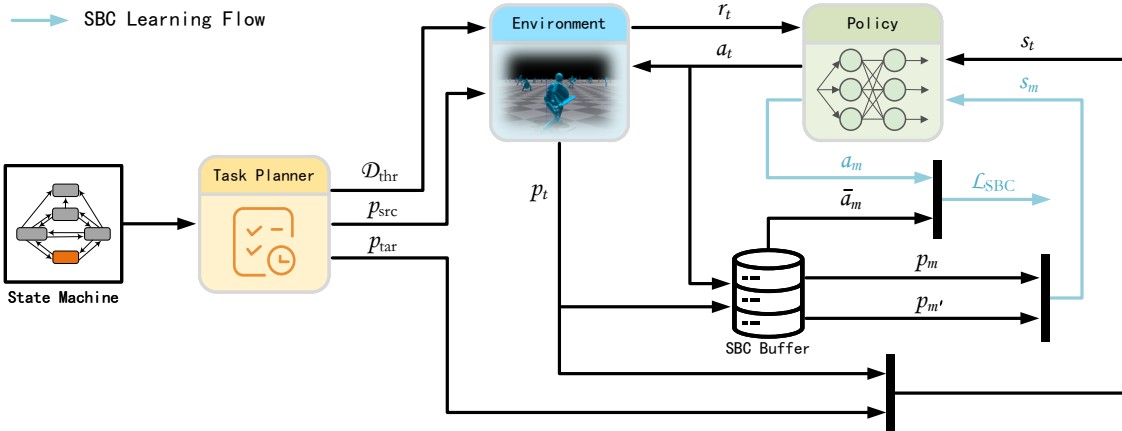

**Figure 2: An overview of our framework on the interactive transition task. With the animation system (state machine) defined by animator, the task planner unit plans the transition task containing source pose $p_{src}$, target pose $p_{tar}$, and deviation threshold $D_{thr}$, so as to carry out pure objective RL. The SBC buffer records the sequence of poses and actions, and produces retargeted states $s_m$ and actions $\bar{a}_m$ for SBC. The SBC learning flow is in blue.**

## 2.3 Motion In-Betweening

Compared with motion transition, motion in-betweening leveraging *Deep Learning* (DL) is more focused on generating rational transition between poses that are far apart. Harvey et al. [8] employ an Encoder-Recurrent-Decoder (ERD) architecture, which uses latent representations to encode the states, as well as a recurrent Long-Short-Term-Memory (LSTM) network to generates them sequentially. Tang et al. [28, 29] adopt ERD architecture as well as CVAE to produce real-time motion transitions. Another solution of in-betweening is to cast motion infilling as a temporal inpainting task and employ a fully-convolutional auto-encoder architecture [9, 11], which can achieve higher computational efficiency.

The most essential difference between motion in-betweening and our approach is that we do not have ground truth data of the missing motion. All these approaches employ ground truth motion sequences to carry out supervised training. Nevertheless, the ideal result of interactive transition is most likely an *Out-of-Distribution* sample for the training data. On the other hand, our approach sets up any possible transition as the tasks for the unsupervised reinforcement learning. Thus, all the animator needs to do is provide the motion clips for transition, and do not require an extensive training set that resembles them.

## 2.4 Physics-Based Character Control

Physics-based control can ensure physical validity and interactions with the environment. This approach has gained popularity since Peng et al. [21] introduced deep reinforcement learning (Deep RL) for physics-based characters control. In their recent and impressive work [22], they employ adversarial imitation reward to learn the latent representations of skills from the motion data and then train a high-level policy that exerts the skills to accomplish several tasks. However, they did not study the tasks of creating transitions between two motion clips.

Several physics-based works study in-betweening, Li et al. [17] use a kinematics-based pretrained in-betweening network [8] to infill the motion sequence and then train a control policy to reproduce it on the simulated humanoid character; Gopinath et al.[7] directly reproduce the ground truth motion clips and outperform their previous work [34] when dealing with unseen motions(in-distribution). The shortcomings of these methods in handling interactive motion transition are consistent with those of kinematics-based in-betweening. In addition, the training conditions in our approach can be adjusted arbitrarily, thereby adapting to the character physical properties of the motion clips for transition, and are not constrained by the physical properties of the training set used in the above frameworks.

## 3 Approach

The interactive motion transition task can be formulated as: given a humanoid character making source pose $p_{src}$, we want the control policy to drive the model by generating torques on the joints and reaching target pose $p_{tar}$. Crucially, the framework does not require ground truth motion data to supervise the training or to imitate; instead, it employs pure objective Reinforcement Learning with only the motion clips that require transition.

Figure 2 provides a schematic overview of the framework. The animator should define the transition as a state machine, which assigns the range of transition moments, possible transition groups, and their rotation range. A task planner unit plans the transition task for the agent in environment, including the source pose $p_{src}$ and the target pose $p_{tar}$ sampled from the state machine, as well as the deviation threshold $D_{thr}$ controlling the difficulty of reaching $p_{tar}$. The environment then reset the agent with $p_{src}$ and judge whether it reached $p_{tar}$ through $D_{thr}$, to carry out RL. Meanwhile, the trajectories of action $a_t$ and pose $p_t$ are recorded by the SBC buffer. During the SBC process, the policy model will be enhanced with the samples from it. Specifically, for a certain pose $p_m$ sampled

from the buffer, we "retarget" its target pose to a subsequent pose $p_{m'}$ to form state $s_m$, which will serve as the expert demonstration in SBC along with the corresponding action $\bar{a}_m$.

## 3.1 Pure Objective RL for Transition

In our framework, motion transition is formulated as a Markov decision process. The humanoid agent starts at the source pose $p_{\text{src}}$, and interacts with a physics simulator environment by taking action $a_t$ and receiving reward $r_t$, according to a policy $\pi(a_t|s_t)$ conditioned on state $s_t$. The policy is trained using proximal-policy optimization (PPO) [25]. The reward we define not only encourages the agent to reach $p_{\text{tar}}$, but also ensures the naturalness of the movement to a certain extent. Here, we detail the state, action and reward of our RL process.

*3.1.1 State.* The state $s_t \triangleq (\hat{p}_t, \hat{p}_{\text{tar}})$ consists of two parts: localized current pose representation $\hat{p}_t$ and localized target pose representation $\hat{p}_{\text{tar}}$. For the humanoid skeleton, our representation of the pose at frame $t$ is similar to the state used by [22], which is $\hat{p}_t \triangleq (R_t, q_t, \dot{q}_t, k_t)$, where $R_t$ is the root translation, $q_t \in \mathbb{R}^{78}$ is the local rotation of all joints using the 6D normal-tangent encoding, $\dot{q}_t \in \mathbb{R}^{31}$ is the local velocity of all joints using a 3D exponential map, and $k_t \in \mathbb{R}^{6 \times 3}$ is the 3D position of 6 key bodies, i.e., the end-effectors [23]. Root translation includes height $R_t^{\text{h}}$, linear velocity $R_t^{\dot{\text{p}}} \in \mathbb{R}^3$, orientation $R_t^{\text{q}} \in \mathbb{R}^6$, and angular velocity $R_t^{\dot{\text{q}}} \in \mathbb{R}^3$. All the configurations are represented with respect to the current location and facing transformation(i.e., "localized"). $\hat{p}_{\text{tar}}$ employs the representation consistent with $\hat{p}_t$, except that it is represented with respect to the transformation of $\hat{p}_t$, and the root height is replaced with integral 3D position $R_t^{\text{p}} \in \mathbb{R}^3$. In contrast with [22], our root orientation is also localized, and only the pitch angle information is retained in $\hat{p}_t$, since the serial transition tasks in our framework will substantially change the character's orientation.

*3.1.2 Action.* The action $a_t$ is the target rotation for all the joints, and that of the spherical joints are encoded using a 3D exponential map, which is consistent with [23]. The proportional derivative(PD) controllers [27] on joints will generate torques that are proportional to the current position error.

*3.1.3 Reward.* If agent is only rewarded when it reaches the target pose, it will be a sparse reward, causing the agent to learn slowly or even unable to learn. Previous work [22] used a decreasing function of the deviation as the reward for location task, while we use the reduction of the current deviation relative to the previous frame, thus avoiding the reward value being affected by the size of the deviation value. Specifically, to encourage the approaching of the target pose at frame $t$, we propose deviation improvement reward $r_t^{\text{imp}}$:

$$r_t^{\text{imp}} = 2.4 I_t^{\text{rp}} + 0.1 I_t^{\text{rv}} + 0.1 I_t^{\text{rq}} + 0.3 I_t^{\text{kp}} + 1.6 I_t^{\text{uq}} \quad (1)$$

We assign higher weights for root position improvement $I_t^{\text{rp}}$, upper body joints rotation improvement $I_t^{\text{uq}}$ and key body position improvement $I_t^{\text{kp}}$, since these deviations are quite visually obvious, and we also find that they have the strongest guiding effect on transition. Root velocity improvement $I_t^{\text{rv}}$ and root rotation improvement $I_t^{\text{rq}}$ have relatively lower weights. We did not adopt

the multiplicative reward function used in [34], since we allow the policy to reduce various deviations in different orders. The linear deviation terms $d_t^{\text{rp}}$, $d_t^{\text{rv}}$ and $d_t^{\text{kp}}$ are computed with the L2 norm in meters, the root rotation deviation $d_t^{\text{rq}}$ computes the scalar rotation of a quaternion about its axis in radians(denoted with $||q||$), while the joint rotation deviation computes the L1 norm of 3D exponential map, as follows:

$$I_t^{\text{rp}} = d_{t-1}^{\text{rp}} - d_t^{\text{rp}} = ||R_{t-1}^{\text{p}} - R_{\text{tar}}^{\text{p}}||_2 - ||R_t^{\text{p}} - R_{\text{tar}}^{\text{p}}||_2 \quad (2)$$

$$I_t^{\text{rv}} = d_{t-1}^{\text{rv}} - d_t^{\text{rv}} = ||R_{t-1}^{\dot{\text{p}}} - R_{\text{tar}}^{\dot{\text{p}}}||_2 - ||R_t^{\dot{\text{p}}} - R_{\text{tar}}^{\dot{\text{p}}}||_2 \quad (3)$$

$$I_t^{\text{rq}} = d_{t-1}^{\text{rq}} - d_t^{\text{rq}} = ||R_{t-1}^{\text{q}} \ominus R_{\text{tar}}^{\text{q}}|| - ||R_t^{\text{q}} \ominus R_{\text{tar}}^{\text{q}}|| \quad (4)$$

$$I_t^{\text{kp}} = d_{t-1}^{\text{kp}} - d_t^{\text{kp}} = \sum_e (||k_{t-1}^e - k_{\text{tar}}^e||_2 - ||k_t^e - k_{\text{tar}}^e||_2) \quad (5)$$

$$I_t^{\text{uq}} = \sum_u w_u S(||q_{t-1}^u - q_{\text{tar}}^u||_1, ||q_t^u - q_{\text{tar}}^u||_1) \quad (6)$$

Here, $k_t^e$ is the position of $e$th key body, $e \in$[left foot, right foot, left hand, right hand, sword, shield], consistent with [22]. $q1 \ominus q2$ denotes the quaternion difference, and $q_t^u$ is the local rotation of $u$th upper body joint, $u \in$[abdomen, neck, right shoulder, right elbow, right hand, left shoulder, left elbow]. $R_{\text{tar}}^{\text{p}}$, $R_{\text{tar}}^{\dot{\text{p}}}$, $R_{\text{tar}}^{\text{q}}$, $k_{\text{tar}}^e$ and $q_{\text{tar}}^u$ represent the parameters of the target pose. $w_u$ is the weight for the rotation deviation of $u$th joint, this design is based on the consideration of the error accumulation problem [19], and we assign higher weight to joints with smaller kinematic chain lengths $c_u$ to the root joints (abdomen, left hip, and right hip):

$$w_u = 1.5^{3-c_u} \quad (7)$$

The design of the negative scaling function $S(x, y)$ is based on our assumption about transition movements: the upper body has less impact on maintaining body balance; thus, it only needs to transit "monotonically" to the target pose. Therefore, we penalize the negative terms in $I_t^{\text{uq}}$ to reduce redundant rotation of the upper body joints:

$$S(x, y) = \begin{cases} x - y & x \geq y \\ 2(x - y) & x < y \end{cases} \quad (8)$$

On the contrary, for joints in the lower body, we hope that they will complete the transition task on the basis of maintaining body balance, rather than directly reducing the lower body joints' local rotation deviation $d_t^{\text{lq}}$ to match the target. In the same way, for the joint angular velocity deviation $d_t^{\dot{\text{q}}}$, we also encourage policy to approach the target in a "roundabout" way, since in some human movements (e.g. walk and run), joint velocity oscillate periodically. Therefore, we include them in the end reward $r_t^{\text{end}}$, which only takes effect at the frame that ends the trajectory, i.e., when falling or a timeout occurs (failure), or when the character reaches the target pose (success):

$$r_t^{\text{end}} = \begin{cases} -3 & \text{failure} \\ 3 + \exp(-4d_t^{\text{lq}}) + \exp(-0.2d_t^{\dot{\text{q}}}) & \text{success} \\ 0 & \text{otherwise} \end{cases} \quad (9)$$

In the same way as Eq. 6, The deviations can be formulated as:

$$d_t^{\text{lq}} = \sum_l w_l ||q_t^l - q_{\text{tar}}^l||_1 \tag{10}$$

$$d_t^{\dot{\text{q}}} = \sum_j w_j ||\dot{q}_t^j - \dot{q}_{\text{tar}}^j||_1 \tag{11}$$

where $w_l$, $w_j$ are the weights of $lth$(lower body joints only) and $jth$ joint respectively, consistent with Eq. 7. $q_t^l$, $\dot{q}_t^j$ are current parameter terms, and $q_{\text{tar}}^l$, $\dot{q}_{\text{tar}}^j$ are target parameter terms. We find that this separate treatment does result in more natural lower body movements.

In early experiments, we found that the trained policy often aligned to the target pose in a high-frequency tremble manner, a phenomenon that can lead to severe visual artifacts. Therefore, we propose tremble reward $r_t^{\text{trem}}$ to penalize joints whose velocities has opposite signs to those of the last frame:

$$r_t^{\text{trem}} = \begin{cases} \sum_{j'} w_{\text{trem}} O(\dot{q}_t^{j'}, \dot{q}_{t-1}^{j'}) & C > C_{\text{thr}} \\ 0 & C \leq C_{\text{thr}} \end{cases}$$

$$C = \sum_{t'=1}^{t} \sum_{j'} O(\dot{q}_{t'}^{j'}, \dot{q}_{t'-1}^{j'}) \tag{12}$$

$$O(x, y) = \begin{cases} 1 & xy < 0 \\ 0 & xy \geq 0 \end{cases}$$

Here, $\dot{q}_t^{j'}$ denotes the velocity of $j'th$ joint Degrees-of-Freedom (DOF, i.e. each dimension of the 3D exponential map), and $w_{\text{trem}} = -0.05$ is the penalization weight. Because a certain amount of joint velocity direction changes would have occurred during the movement, we compute the cumulative value of changes $C$, and only penalize $r_t^{\text{trem}}$ when $C$ exceeds the threshold $C_{\text{thr}}$, which is set to 100.

Finally, we get the complete reward $r_t$ by accumulating the above terms:

$$r_t = r_t^{\text{imp}} + r_t^{\text{end}} + r_t^{\text{trem}} \tag{13}$$

## 3.2 Task Planning

### 3.2.1 Pose Sampling.
During training, we randomly sample the transition task from the state machine, including the source motion, target motion, source motion transition moment, and target motion rotation. Each motion clip has a start time point and a cancel time range (usually the last part of the motion clips, i.e., the recovery motion) specified by the animator. The former is used as the transition moment when motion serves as target motion, while the latter is used as the uniform sampling range for the transition moment when motion serves as source motion. With these two transition moments, we acquire the source pose $p_{\text{src}}$ and the target pose $p_{\text{tar}}$ from motion clips. The position, velocity, and orientation configurations of $p_{\text{tar}}$ will be rotated (around the z-axis) by the rotation angle uniformly sampled from a particular range. After that, the rotated root position of the target pose will add the offset $R_{\text{offset}}^{\text{p}}$ produced by the source pose:

$$R_{\text{offset}}^{\text{p}} = \tilde{R}_{\text{offset}}^{\text{p}} - \frac{\tilde{R}_{\text{offset}}^{\text{p}} \cdot \mathbf{u}_{\text{up}}}{||\mathbf{u}_{\text{up}}||_2}$$

$$\tilde{R}_{\text{offset}}^{\text{p}} = R_{\text{src}}^{\text{p}} + (R_{\text{src}}^{\dot{\text{p}}} + R_{\text{end}}^{\dot{\text{p}}})T/2 \tag{14}$$

where $R_{\text{src}}^{\text{p}}$ is the root position of source pose, and $\mathbf{u}_{\text{up}}$ is the up-vector direction that is perpendicular to the ground. We assume that the root linear velocity of source pose $R_{\text{src}}^{\dot{\text{p}}}$ changes uniformly to $R_{\text{end}}^{\dot{\text{p}}}$ over time $T$ ($T = 0.8$) in the transition, from which we estimate the root position offset. Compared with [4], we simply set $R_{\text{end}}^{\dot{\text{p}}}$ to zero, since in most motion clips the root velocity at start time point is close to still.

### 3.2.2 Curriculum Learning.
To judge whether the character reached the target pose, we employ deviation threshold $D_{\text{thr}} \triangleq (d_{\text{thr}}^{\text{rp}}, d_{\text{thr}}^{\text{kp}}, d_{\text{thr}}^{\text{q}})$, where $d_{\text{thr}}^{\text{rp}}$, $d_{\text{thr}}^{\text{kp}}$ and $d_{\text{thr}}^{\text{q}}$ are the threshold of $d_t^{\text{rp}}$ (Eq. 2), $d_t^{\text{kp}}$ (Eq. 5), and joints local rotation deviation $d_t^{\text{q}}$ respectively. The calculation of $d_t^{\text{q}}$ is similar to Eq. 10, except that it includes all joints:

$$d_t^{\text{q}} = \sum_j w_j ||q_t^j - q_{\text{tar}}^j||_1 \tag{15}$$

The criterion for reaching the target pose is $(d_{\text{thr}}^{\text{rp}} > d_t^{\text{rp}}) \wedge (d_{\text{thr}}^{\text{kp}} > d_t^{\text{kp}}) \wedge (d_{\text{thr}}^{\text{q}} > d_t^{\text{q}})$, i.e. all the deviation terms are less than the threshold terms. In addition to this, there are two situations of termination: falling (body parts other than feet contact the ground) and timeout (agent steps more than 60), which are considered failures.

To accelerate training, we adopt a curriculum learning strategy with respect to the difficulty of reaching the target pose. Specifically, we compute the achievement rate of tasks through a sliding average window of length 100, once the task achievement rate exceeds 88%, it means that the current difficulty is too easy for the policy. Therefore, we replace the deviation thresholds with new values $\tilde{d}_{\text{thr}}^{\text{rp}}, \tilde{d}_{\text{thr}}^{\text{kp}}, \tilde{d}_{\text{thr}}^{\text{q}}$:

$$\tilde{d}_{\text{thr}}^{\text{rp}} = \max(0.9d_{\text{thr}}^{\text{rp}}, 0.01)$$

$$\tilde{d}_{\text{thr}}^{\text{kp}} = \max(0.9d_{\text{thr}}^{\text{kp}}, 0.09) \tag{16}$$

$$\tilde{d}_{\text{thr}}^{\text{q}} = \max(0.9d_{\text{thr}}^{\text{q}}, 0.15)$$

We employ an exponential form to reduce thresholds, thus as higher and higher accuracy is achieved, the thresholds decrease more and more slowly until they all reach their constant lower bounds (i.e., final accuracy). The initial values of $d_{\text{thr}}^{\text{rp}}$, $d_{\text{thr}}^{\text{kp}}$ and $d_{\text{thr}}^{\text{q}}$ are 0.3, 3 and 1 respectively. We find this curriculum learning strategy to be a very significant facilitator of the training process. If the policy is trained directly with the final accuracy, the task achievement rate almost remains zero throughout the training.

We also propose a stay probability $P_{\text{stay}} = 0.5$ to increase the number of times the policy handles complex tasks. When the agent fails on a task, the task will be retained with $P_{\text{stay}}$ chance instead of randomly sampling another. This measure can further increase the training speed.

## 3.3 Self-Behavior Cloning

The idea of Self-Behavior Cloning (SBC) is similar to Hindsight Experience Replay (HER) [1]. We hope that the policy can not only learn from the experience of accurately achieving the task, but also learn from the experience of failure, which can be viewed as achieving similar tasks. Given the state $s_t \triangleq (\hat{p}_t, \hat{p}_{\text{tar}})$, the generated action $a_t$ may not be the proper choice for the transition to pose $p_{\text{tar}}$, but it is the right choice for transitioning to pose $p_{t+1}$,

---

**ALGORITHM 1:** Training with SBC

---

$\pi \leftarrow$ initialize policy;
$V \leftarrow$ initialize value function;
$\mathcal{B} \leftarrow \emptyset$ initialize replay buffer;
$\mathcal{B}_{\text{SBC}} \leftarrow \emptyset$ initialize SBC buffer;

**while** *not done* **do**
    **for** *environment* $i \leftarrow 1$ **to** $m$ **do**
        $\tau_i \leftarrow \{(\hat{p}_t, \hat{p}_{\text{tar}}), a_t, r_t\}_{t=0}^{T-1}$ collect trajectory with $\pi$;
        **for** *time step* $t \leftarrow 0$ **to** $T-1$ **do**
            **for** $k \leftarrow t+1$ **to** $T-1$ **do**
                **if** $(p_t, p_k)$ *is valid* **then**
                    $\hat{p}_k \leftarrow$ localize $p_k$ with respect to $p_t$;
                    store$\{(\hat{p}_t, \hat{p}_k), a_t\}$ in $\mathcal{B}_{\text{SBC}}$;
                **end**
            **end**
        **end**
        store $\tau_i$ in $\mathcal{B}$;
    **end**
    **for** *update step* $t \leftarrow 1$ **to** $n$ **do**
        $b_{\text{SBC}} \leftarrow$ sample $K$ demonstrations $\{s_j, a_j\}_{j=1}^{K}$ from $\mathcal{B}_{\text{SBC}}$;
        update $\pi$ according to Eq. 17 using $b_{\text{SBC}}$;
        update $\pi$ and $V$ using data from $\mathcal{B}$;
    **end**
**end**

---

$p_{t+2}$, as well as all poses before the end of the trajectory. Unlike HER, which adds the reassembled failed trajectories to the replay buffer, we directly reuse the state-action trajectory as the expert demonstrations to perform Behavior Cloning (BC). This is because our reward function is neither sparse nor binary, and recomputing the rewards for all transitions on the failed trajectory is very time-consuming. In addition, updating the policy through supervised learning is more efficient than using rewards.

As Algorithm 1 states, for each time step $t$, we localize the following pose $p_k$ (serves as the target pose) with respect to the current pose $p_t$ (serves as the source pose) to form a transition task, as long as this group is "valid", i.e., $p_t$ and $p_k$ are from a continuous trajectory without being reset midway. The task $(\hat{p}_t, \hat{p}_k)$ as well as the action $a_t$ form an expert demonstration, which will be stored in the SBC buffer. During the update phase, an expert demonstrations batch $b_{\text{SBC}}$ is randomly sampled from the SBC buffer; the policy objective is then given by:

$$\arg\min_{\pi} \sum_{(s_i, a_i) \sim b_{\text{SBC}}} w_{\text{SBC}} \frac{T-1}{T-g_i} ||a_i - \pi(s_i)||_2^2 \qquad (17)$$

Here, $g_i$ is the time step gap between the source pose and target pose in $s_i$, used for balancing the impacts of demonstrations with different gap lengths since those with shorter lengths are more numerous. Within each update step, SBC and pure objective RL will be executed sequentially, and $w_{\text{SBC}}$ is the coefficient to balance their impacts, which is set to 0.533.

During the exploration of an agent, plenty of falling motions have low correlation with the correct motions expected in the tasks. Thus, the Behavior Cloning of these demonstrations is harmful. To ensure the relevance of the demonstrations to the tasks, we add

considerations of reward in the validity determination, i.e., the root position improvement $I_t^{\text{rp}}$ (Eq. 2) of the demonstration must be positive.

## 4 Experiments

### 4.1 System Setup

*4.1.1 State Machine.* Our 3D articulated humanoid character model is equipped with a sword and shield and has 31 DOF, which is consistent with [22]. The motion clips used to construct the state machine come from the sword&shield stunts dataset from [22] and have already been retargeted to the character model by them. We choose 9 motion clips, including common attack and movement motions, and assign different ranges of transition moments for them with complete coverage of their recovery motions (periodic motions such as run have a range of almost their entire duration). The state machine is a fully connected graph (including self-loop). Thus, there are a total of 81 possible groups of transition. The rotation range of target motion is uniformly set to $(-\frac{\pi}{4}, \frac{\pi}{4})$.

*4.1.2 Training.* The character is simulated in Isaac Gym [18], with a simulation frequency of 120Hz and policy execution frequency of 30Hz. Both the policy network and value function network are multi-layer perceptions using independent state representation and are implemented with PyTorch [20]. The policy is trained for 60000 iterations on a single NVIDIA 3090 GPU with 4096 environments simulated in parallel, taking about 1.8 days, and yields approximately 4 billion samples, corresponding to about 4 years of simulated time.

### 4.2 Comparison to Existing Method

We compare our proposed method with the existing mainstream method, i.e., the interpolation-based method. We use an ease-in-ease-out cubic curve to blend two motion clips, and applied time warping to their blended parts. The average time steps of transition in our method is used as the fixed duration for blending (12 frames). Figure 3 provides a qualitative comparison of the two methods. The interpolation-based method suffers from severe visual artifact of foot-skating, while our method completely avoids it. In addition, since the interpolation-based method uses a fixed transition duration, the angular and linear velocity during transition are determined by the deviation between the source pose and the target pose, which will lead to abrupt velocity change in many cases. Our method, on the other hand, determines the transition duration based on the difficulty, and better follows the velocity of the source pose and the target pose.

### 4.3 Evaluation Metrics

We use the well-trained policy to execute all possible transition tasks sampled from the state machine and evaluate the effectiveness of our framework with metrics in four aspects: accuracy, naturalness & rapidity, robustness, and training efficiency. The average accumulated value of $r_t$ (i.e., Return) can be regarded as the synthesis of the above metrics. The evaluation sampling intervals of transition moment and rotation are 0.03$s$ and $\frac{\pi}{180}$, respectively, with a total of 460278 different transition tasks.

**Table 1: Quantitative result of our method, as well as the ablation study on SBC, URP, and TR refer to Self-Behavior Cloning, upper body redundancy penalization, and tremble reward, respectively.**

| Methods | FRPD↓ | FRVD↓ | FRRD↓ | FKPD↓ | FJRD↓ | FJAD↓ | SURI↑ | TC↓ | TS↓ | MS↑ | TTAR↑ | IRFA↓ | Return↑ |
|---|---|---|---|---|---|---|---|---|---|---|---|---|---|
| Ours | 0.0038 | 0.6556 | 0.1677 | 0.0513 | 0.1071 | 3.2404 | -0.1677 | 71.90 | 12.02 | $e^{-3.7}$ | 1.000 | 10362 | 5.4027 |
| No SBC | 0.0306 | 0.6741 | 0.2080 | 0.1585 | 0.1484 | 2.5384 | -0.1685 | 76.15 | 12.99 | × | 0.889 | 14809 | 4.5615 |
| No URP | 0.0237 | 0.6906 | 0.1699 | 0.0903 | 0.1284 | 3.3104 | **-0.3873** | 81.20 | 14.09 | × | 0.889 | 13157 | 4.1353 |
| No TR | 0.0048 | 0.7548 | 0.1045 | 0.0613 | 0.0592 | 1.2915 | 0.0482 | **154.27** | 10.23 | $e^{-3.5}$ | 1.000 | 3010 | 2.8425 |

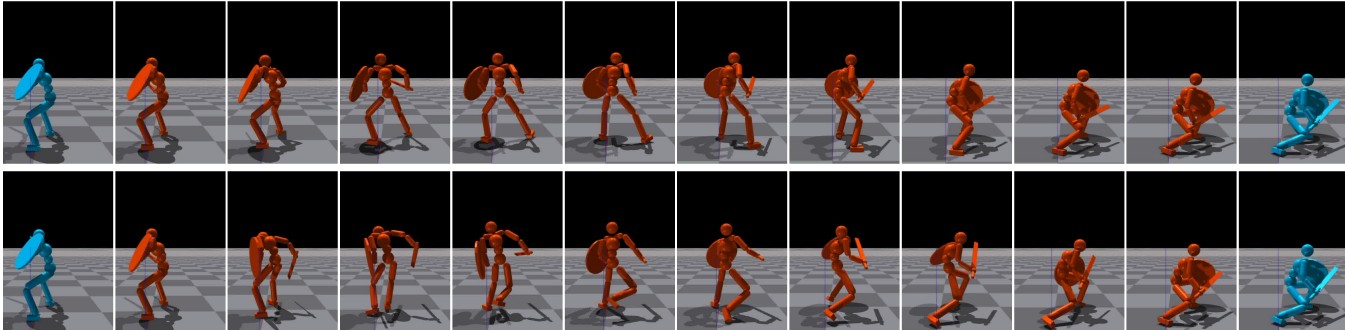

**Figure 3: Qualitative comparison between the interpolation-based method and our method when dealing with a same transition task, shown in the upper and lower rows of the figure respectively. There is a large rotation angle between the source pose and the target pose, thus the problems of foot-skating and velocity fluctuation in the interpolation-based method are very serious, and our method avoids them well.**

*4.3.1 Accuracy.* We use the deviation terms $d_t^{\mathrm{rp}}$ (Eq. 2), $d_t^{\mathrm{rv}}$ (Eq. 3), $d_t^{\mathrm{rq}}$ (Eq. 4), $d_t^{\mathrm{kp}}$ (Eq. 5), $d_t^{\mathrm{q}}$ (Eq. 15) and $d_t^{\dot{\mathrm{q}}}$ (Eq. 11) at the final frame to evaluate the accuracy of a certain transition, and the average values of them across all tasks will be used as the accuracy metrics: final root position deviation (FRPD), final root velocity deviation (FRVD), final root rotation deviation (FRRD), final key body position deviation (FKPD), final joint rotation deviation (FJRD), and final joint angular velocity deviation (FJAD). For all of them, lower is better.

*4.3.2 Naturalness & Rapidity.* We evaluate the naturalness of transition by unsupervised metrics, including scaled upper body joints rotation improvement (SURI) being the average value of $I_t^{\mathrm{uq}}$ (Eq. 6) and tremble count (TC) being the average value of $C$ (Eq. 12). Also, due to the need to respond to user inputs, our framework should generate sufficiently rapid transition motions, which is evaluated by the average time steps of transition (TS). For SURI, higher is better, since it additionally penalizes the upper body movement away from the target pose. For both TC and TS, lower is better.

*4.3.3 Robustness.* The policy is modeled by a multivariate normal distribution over the action space, with a state-dependent mean $\pi(s_t)$ and a fixed diagonal covariance matrix $\Sigma = \mathrm{diag}(\sigma, \sigma, ...)$. The value of $\sigma$ controls the amount of noise applied to the joint torque, which is set to $e^{-2.9}$ during training. During operation, too high a $\sigma$ will lead to a decrease in transition stability, and the character will fall during some tasks. In the evaluation, $\sigma$ is set to $e^{-3.7}$, and the total task achievement rate (TTAR) in this condition along with the maximum sigma value (MS) that guarantees 100% TTAR are

used to evaluate the robustness of the policy. For both TTAR and MS, higher is better.

*4.3.4 Training Efficiency.* Our framework employs a curriculum the learning strategy for the transition accuracy and the number of iterations when reaching the final accuracy (IRFA) will be used as the metric to evaluate training efficiency.

### 4.4 Result and Ablation

We evaluate our integral method by the above metrics, as reported in the first row of Table 1. The result shows that our method can achieve all the sampled interactive transition tasks with a certain amount of action noise and achieves extremely high accuracy in several deviation metrics (FRPD, FKPD and FJRD) that users are more sensitive to. When reaching the final accuracy, we find that the policy already has better effects with the deterministic pattern (i.e., directly using the mean as the output). Subsequent iterations mainly improve the stability of the sampling pattern. We also find that the policy can usually handle a more extensive range of rotation than in training, reflecting the strong generalization ability of it.

*4.4.1 Self-Behavior Cloning.* To show the importance of our proposed SBC mechanism, we compare our method with a baseline model training without SBC (No SBC), whose performance is reported in the second row of Table 1. Our method significantly outperforms the baseline on almost all metrics except FJAD. Without SBC, extreme fluctuations occurred in the mean return curves (see Figure 4(a)) during training, and the trained policy can not handle any tasks transitioning to the stab motion, even with the

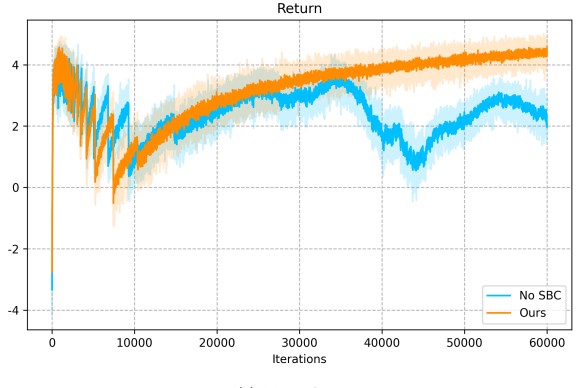

(a) Mean Return

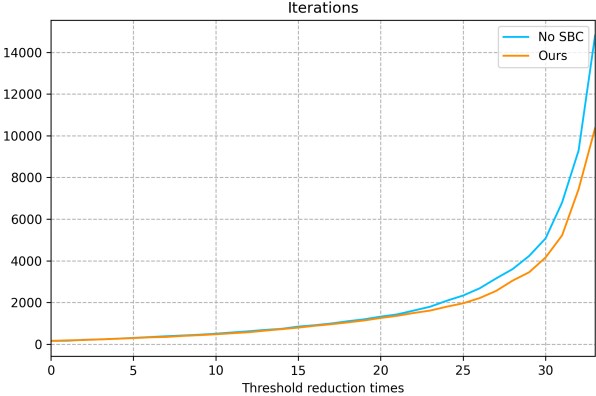

(b) Iterations required for different threshold reduction times

**Figure 4: Curves of mean return and threshold reduction iterations during training.**

deterministic pattern. The SBC mechanism also improves training efficiency; the policy takes less time to reach the same transition difficulty (see Figure 4(b)), and reaches the final accuracy about 4000 iterations earlier.

*4.4.2 Upper Body Redundancy Penalization.* The third row of Table 1 reports the model trained without the additional penalization of the redundant movement on the upper body (No URP). It suffers from the increased redundant movement, causing lower performance on all metrics (especially SURI) and unnatural swing of the upper limbs. The trained policy can also not handle any tasks transitioning to a certain motion, the slash up motion. This confirms our prior "monotonically" hypothesis about upper body movement, which can reduce learning difficulty and improve naturalness.

*4.4.3 Tremble Reward.* The model trained without tremble reward (No TR) achieves higher training efficiency and robustness by exploiting unnatural and unacceptable tremble behaviors (similar to the result of interpolation-based method), and TC is doubled in the evaluation, as shown in the fourth row of Table 1. Empirically, we

find that the efficacy of tremble reward is not sensitive to the values of $C_{thr}$ and $w_{trem}$ in Eq. 12.

## 5 Conclusion and Discussion

In this paper, we propose a novel unsupervised framework for synthesizing physically valid transitions that seamlessly bridge motion clips, enabling natural and interactive character animation with a hybrid pattern. We have shown that our approach can handle any transition task sampled from a complex state machine with high accuracy and naturalness in the evaluation. Notably, our proposed SBC mechanism specialized for RL of motion transition has been proven to promote the training process and final result. In contrast to using purely physical control to synthesize animation, our hybrid manner can be regarded as a compromise between the current mainstream animation system and the emerging physics-based animation, avoiding the instability problem of the latter.

Despite its remarkable advantages, we note a few shortcomings. The well-trained policy tends to sacrifice the accuracy of joint angular velocity (reflected in FJAD) to improve the accuracy of other aspects. A future research avenue could involve a user study to evaluate the visual prominence of different transition deviations, thereby improving the design of reward and curriculum learning. Besides, our method takes on average 0.7 ms to inference an action, however the simulation of a frame takes on average 14 ms. We plan to investigate whether the efficiency is sufficient for real-time applications with other physics engines. In addition, as we mainly focus on the policy's adaptability to various transitions, estimating of the target pose root position offset in our framework is relatively simple (see Eq. 14), and redundant appel movements may occur in some transitions. The animator can employ more complex computations of the offset to obtain the desired result. Furthermore, without the information from a reference motion dataset, our approach sometimes produces unnatural movements on the upper body. We are interested in exploring other unsupervised methods to mitigate these artifacts, such as integrating energy efficiency objectives [35] into our reward or employing a musculoskeletal model to generate more biomechanically plausible motion [16].

Another application of hybrid character animation is generating dynamic responses to unexpected impacts [26, 36]. We are interested in extending our approach for such application, since it will be very suitable for controlling the character to recover to a motion clip after being impacted, which is difficult to achieve with purely physical control in previous works.

## Acknowledgments

We would like to thank the anonymous reviewers for their feedback, Zhenjiang Du, Feng Tian, and Sophyani Banaamwini Yussif for their support for this work.

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
