# OpenReview forum: "PIMT: Physics-Based Interactive Motion Transition for Hybrid Character Animation"
_acmmm.org/ACMMM/2024/Conference — MM2024 Poster_

### Official Review · Reviewer_kEYm · 2024-05-24

**Rating:** 5
**Confidence:** 2

**Summary:**

The paper presents a Physics-Based Interactive Motion Transition (PIMT) framework for generating seamless and natural character animations in real-time 3D interactive applications. The framework combines motion capture animation with physical simulation to connect motion clips smoothly. Key innovations include:
1.Self-Behavior Cloning (SBC): Enhances unsupervised reinforcement learning for motion transitions, leveraging exploration trajectories to improve the learning process.
2.Sophisticated Reward Function: Integrates objectives for transition accuracy and naturalness, balancing between diversity and stability.
3.Task Planning and Curriculum Learning: Employs a state machine and a curriculum learning strategy to plan and train the control policy efficiently.

**Strengths:**

The framework allows arbitrary specification of transition moments and target motion rotations, providing high responsiveness and user control. Unlike interpolation methods, the physics-based approach ensures that transitions adhere to human body dynamics, avoiding visual artifacts like foot-skating.

**Limitations:**

Without reference motion datasets, the method sometimes produces unnatural upper body movements.  The framework's inference and simulation times may not be sufficient for all real-time applications, and further efficiency improvements are needed.

**Suitability:**

3

---

### Official Review · Reviewer_BUgw · 2024-05-24

**Rating:** 5
**Confidence:** 3

**Summary:**

This paper proposes a novel framework for synthesizing physically valid motion transitions, enabling natural and interactive character animation with a hybrid pattern. Different from other in-betweening approaches, this method does not require ground truth data on the missing motion for supervised learning.  Additionally, this approach proposes complex reward functions and Self-Behavior Cloning to learn the transition policy, improving the transition accuracy,

Generally, I believe there are a lot of good design decisions in the paper for unsupervised physics-based interactive motion transitions. The curriculum learning strategy and Self-Behavior Cloning also help to improve the policy learning process. The main concern I have about this approach is whether the model can learn well on diverse datasets.

**Strengths:**

The method incorporates the physical simulation in motion in-between, eliminating the common artifacts such as foot skating artifacts.

The method is tailored for practical usage where there is no ground truth data in game creation pipeline.

The method incorporates carefully designs to keep continuous transition between physical simulation and key-frames.

**Limitations:**

The method designs a complex reward function and has only been trained and tested on one dataset. I'm wondering how well the approach generalizes and whether it can be applied to more complex interactive data. Can it generalize to arbitrary data without training?

The synthesized motion is too short in video, making it difficult to evaluating the motion naturalness.

Despite this paper conducting exhaustive ablation studies, it lacks comparisons between recent in-between motions.

Leveraging the average duration of transition (TS) for naturalness may not be appropriate; a lower transition duration has no relationship to motion naturalness. Conversely, too low duration might indicate a physical-plausible but unnatural motion.

Diversity is commonly evaluated by the difference between synthesized results rather than the hyper-parameters of a model. The range of a hyper-parmamter (sigma) cannot indicate the diversity of generation.

**Suitability:**

2

---

### Official Review · Reviewer_HcPG · 2024-05-31

**Rating:** 4
**Confidence:** 3

**Summary:**

This paper investigates motion transitions task by introducing a framework to produce hybrid character animation, which combines motion capture animation and physical simulation animation that seamlessly connects the front and back motion clips. In order to avoid the limitations of interpolation which may produce severe visual artifacts, this work uses physics-based character control technique to bridge the motion clips seamlessly. Besides, the proposed Self-Behavior Cloning (SBC) mechanism to help the policybetter utilize its exploration trajectories, and provides animators with greater creative freedom.

**Strengths:**

The technique is sound. This paper extend physics-based character control into  interactive motion transition to improve the quality of motion transition via reducing visual artifacts. Furthermore, the proposed SBC method could capture richer knowledge of motion transition in the exploration process. In addition, the details of the proposed technique, e.g., task planning, self-behavior cloning are presented clearly.

**Limitations:**

Lack of detailed analysis of experimental results. This paper provides performance comparison of the proposed method with several baselines, and give the results of ablation experiment. However, the in-depth analysis of the experimental results is insufficient, especially for the principle analysis of the method.

**Suitability:**

3

---

### Official Review · Reviewer_JLXx · 2024-06-01

**Rating:** 3
**Confidence:** 2

**Summary:**

The paper addresses the problem of motion in-betweening between two sequences of character animations. The method used is a hybrid approach combining physics constraints for validity and interpolation for overall smoothness and diversity. The hybrid framework allows for an interactive selection of transition frames.

**Strengths:**

- The motivation for interactive control over the transition and resulting generation is well motivated. The proposed hybrid scheme is certainly a valid approach towards the problem.

- No requirement of supervision for the motion transfer task, relying upon hand crafted objectives in a RL setting.

**Limitations:**

- The writing and presentation for the paper could be improved upon. For example, Fig-4b is not entirely clear to interpret and perhaps a different visualization could help. In Fig-2, the overall pipeline seems readble but also a bit confusing on first glance (other experienced readers might differ in this opinion).

- The experiment section does not include any comparisons with related methods and only compares with interpolation baseline. So although the ablation results show
the motivation behind the design choices, the evaluation is not clear enough to decide upon the proposed contributions.

**Suitability:**

2

---

### Meta-Review · Area_Chair_EECk · 2024-06-26

**Recommendation:** Accept (Poster)
**Confidence:** 4

**Metareview:**

Most of the reviewers made positive ratings regarding the submission, the authors are encouraged to take their constructive comments into the final version.